# Treatment Sequencing Strategies in Advanced Neuroendocrine Tumors: A Review

**DOI:** 10.3390/cancers14215248

**Published:** 2022-10-26

**Authors:** Aman Chauhan, Jaydira Del Rivero, Robert A. Ramirez, Heloisa P. Soares, Daneng Li

**Affiliations:** 1Division of Medical Oncology, Department of Internal Medicine, Markey Cancer Center, University of Kentucky, Lexington, KY 40536, USA; 2Developmental Therapeutics Branch, National Cancer Institute, NIH, Bethesda, MD 20892, USA; 3Division of Hematology Oncology, Vanderbilt University Medical Center, Nashville, TN 37232, USA; 4Division of Medical Oncology, Department of Internal Medicine, Huntsman Cancer Institute at University of Utah, Salt Lake City, UT 84112, USA; 5Department of Medical Oncology & Therapeutics Research, City of Hope Comprehensive Cancer Center, Duarte, CA 91010, USA

**Keywords:** neuroendocrine tumors (NET), peptide receptor radionuclide therapy (PRRT), treatment sequencing, efficacy, safety, clinical trials

## Abstract

**Simple Summary:**

Neuroendocrine tumors (NETs) have become increasingly common. There are several effective treatment options for advanced NETs. However, there are limited clinical trial data and published practical information on how these different treatments should be sequenced. This review assesses randomized, controlled clinical trial data in advanced NETs to provide an expert perspective on treatment sequencing for important clinical scenarios, ranging from local disease to high-volume metastatic NETs. The best practices provided in this review may be useful for clinicians considering treatment options and sequencing for their patients with advanced NETs.

**Abstract:**

Neuroendocrine tumor (NET) incidence has grown. The treatment landscape for advanced NETs is rapidly evolving, but there are limited head-to-head data to guide treatment sequencing decisions. We assessed the available clinical data to aid practicing clinicians in their routine clinical decision-making. Clinical trials have demonstrated efficacy benefits for new therapies in advanced NETs. Emerging long-term data from these trials have enabled clinicians to make more accurate risk-benefit assessments, particularly for patients receiving multiple lines of therapy. However, clinical data specifically regarding treatment sequencing are limited. In lieu of definitive data, treatment sequencing should be based on disease-related factors (e.g., site of tumor origin, volume of disease) and patient-related characteristics (e.g., comorbidities, patient preferences). Clinical decision-making in advanced NETs remains highly individualized and complex; important evidence gaps regarding treatment sequencing remain. Given this, advanced NET management should be a joint effort of multidisciplinary teams at referring and high-volume centers. Additional clinical trial and real-world evidence are needed to meet the challenge of understanding how to sequence available NET therapies. Until these trials are conducted, the best practices provided in this review may serve as a guide for clinicians making treatment sequencing decisions based on the available data.

## 1. Introduction 

Neuroendocrine tumors (NETs) are rare and heterogenous neoplasms with continued rising incidence [1,2]. The indolent nature of most NETs has led to high prevalence with over 170,000 patients with NETs estimated in the United States, making NETs the second most common neoplasms of gastrointestinal origin [1,3].

The systemic treatment landscape for advanced NETs has evolved with the approvals of somatostatin analogs (SSAs; octreotide long-acting repeatable [LAR] and lanreotide), the tyrosine kinase inhibitor sunitinib, the mammalian target of rapamycin (mTOR) inhibitor everolimus, and radioligand therapy (RLT, also referred to as peptide receptor radionuclide therapy) with [^177^Lu]Lu-DOTA-TATE (^177^Lu-DOTATATE; Figure 1) [4,5]. These contemporary therapies can control symptoms and/or delay disease progression [6,7,8,9,10,11]. However, optimal treatment strategies have not been defined due to an absence of head-to-head studies [12,13].

Although surgery is possible for localized NETs and in selected metastatic NETs, many patients are diagnosed with inoperable advanced or metastatic disease requiring medical therapy [14]. SSAs are typically chosen as first-line systemic therapy owing to their antiproliferative effects, capacity to improve carcinoid syndrome, and favorable safety profile [5,6,7,15]. Potential second-line systemic treatment options depend on individual patient circumstances and include everolimus for pancreatic NETs (pNETs) and nonfunctional NETs of gastrointestinal or lung origin [5,9,10,16], sunitinib for pNETs [8,17], ^177^Lu-DOTATATE for somatostatin receptor-positive gastroenteropancreatic NETs (GEP-NETs) [5,11,12,18], and capecitabine-temozolomide (CAPTEM) for certain GEP-NETs [19]. However, customizing the most appropriate treatment program does not always fall neatly into a straightforward linear choice of recommended first-line then later-line therapies [5,12,13,20].

We review the pertinent clinical trial data of patients with advanced NETs to guide clinical decision-making and treatment sequencing.

## 2. Materials and Methods 

This review focuses on the management of well-differentiated advanced GEP-NETs and typical/atypical pulmonary carcinoid tumors [21,22]. Poorly differentiated neuroendocrine carcinoma, including small-cell and large-cell neuroendocrine carcinoma, were tumor types considered out of scope for this review. Medline (via PubMed) was searched for articles indexed as randomized clinical trials and their associated secondary analyses, containing the following terms: octreotide long-acting release, lanreotide, everolimus, sunitinib, ^177^Lu-DOTATATE, and cytotoxic chemotherapy. Included data pertain only to eight randomized, double-blind, controlled Phase III trials (PROMID [6], CLARINET [7], SUNNET [8], RADIANT-2 [23], RADIANT-3 [9], RADIANT-4 [10], NETTER-1 [11], and SPINET [24]) along with one randomized Phase II trial that has changed standard of care over the last 5 years (ECOG E2211) [19]. Streptozocin-based regimens [25,26] are not cited as preferred regimens in clinical guidelines and thus not reviewed herein. Telotristat ethyl, an approved treatment for carcinoid syndrome diarrhea management in SSA-refractory patients based on the TELESTAR study, is also not reviewed herein as it is a symptom management drug rather than an antitumor agent.

## 3. Results

The designs, types of eligible patients, and endpoints assessed differed to varying extents across the nine randomized controlled studies (Figure 2) [6,7,8,9,10,11,19,23,24,27,28,29,30,31]. PROMID was the only study of treatment-naïve patients, although only 16% of the CLARINET population had received prior treatment [7]. In contrast, NETTER-1 enrolled patients who had disease progression on first-line SSA therapy [11]. NETTER-1 (^177^Lu-DOTATATE vs. high-dose octreotide LAR 60 mg every 4 weeks) [11] and ECOG E2211 (capecitabine plus temozolomide vs. temozolomide) [19] were the only trials to use an active comparator as a control. The primary efficacy endpoint was time to tumor progression in PROMID [6] and progression-free survival (PFS) in the other trials [7,8,9,10,11,19,23,24], where the former metric measured time to disease progression or tumor-related death, and the latter measured time to disease progression or death from any cause.

### 3.1. Efficacy

Direct comparison between the trials is not possible because of variations in patient selection and study methodology. For instance, regarding the interventions, SUNNET patients could receive SSAs at the investigator’s discretion [8], RADIANT-2 patients received IM octreotide concomitant with their randomized treatment [23], and NETTER-1 patients on ^177^Lu-DOTATATE also received IM octreotide [11].

Seven of the nine trials showed a benefit in time to tumor progression (PROMID) [6] or PFS (CLARINET [7], SUNNET [8], RADIANT-3 [9], RADIANT-4 [10], NETTER-1 [11], and ECOG E2211 [19]) in favor of the interventional arm relative to the control arm (Table 1). 

The remaining two trials, RADIANT-2 and SPINET, did not show statistically significant PFS benefits for the interventional arm. In the RADIANT-2 study of patients with advanced NETs associated with carcinoid syndrome, the combination of everolimus plus octreotide prolonged PFS versus octreotide alone, but the study primary endpoint was not met as the *p* value narrowly missed the prespecified boundary denoting statistical significance [23]. Nevertheless, the finding provided an initial indication of the potential PFS benefit of everolimus in patients with advanced NETs, which was subsequently confirmed in the RADIANT-3 study of patients with pNETs [9] and RADIANT-4 study of patients with nonfunctional GI or lung NETs [10]. 

The SPINET study (lanreotide vs. placebo) was unique by enrolling patients with somatostatin receptor-positive typical and atypical carcinoid lung NETs [24]. Enrollment was stopped early because of slow accrual, and patients without centrally assessed progression during the double-blind phase transitioned to open-label lanreotide [24]. The primary endpoint was also adapted to centrally assessed PFS during the double-blind and open-label lanreotide phases in patients initially randomized to lanreotide [24]. Median PFS was 16.6 (95% CI, 12.8–21.9) months in the lanreotide randomized group and appeared longer in the subgroup of patients with typical carcinoid lung NETs (21.9 months, 95% CI, 12.8–not calculable) than atypical carcinoid lung NETs (14.1 months, 95% CI, 5.6–16.6) [24]. In the double-blind phase, median PFS for lanreotide and placebo, respectively, was 16.6 versus 13.6 months (HR, 0.90; *p* = 0.769) in the entire population, 21.9 versus 13.9 months in the subgroup with typical carcinoid lung NETs, and 13.8 versus 11.0 months in the subgroup with atypical carcinoid lung NETs [24].

Per protocol exploratory analysis of PROMID [6], CLARINET [7,31], RADIANT-3 [9], RADIANT-4 [10], and NETTER-1 [11] revealed that the extended time to tumor progression or PFS in favor of the interventional arm relative to the control arm occurred irrespective of randomization stratification factors, and predefined demographic and prognostic factors.

Extensive follow-up is required to demonstrate significant gains in overall survival (OS) owing to the often indolent nature of advanced NETs. In five (PROMID [29], SUNNET [27], RADIANT-2 [32], RADIANT-3 [28], and NETTER-1 [30]) of the nine randomized controlled Phase III trials, final OS (observed all-cause) was reported long after the cutoff date for the primary efficacy analysis, during which time multiple factors can have an influence on mortality. One of these factors is in-trial treatment crossover and post-protocol drug therapy, which affects between-group differences in OS. None of the active treatments in these five trials produced a statistically significant prolongation of OS at the time of the most recent analysis although clinically meaningful differences in median OS were observed in the SUNNET and NETTER-1 studies (Table 1). 

In SUNNET, 9 deaths were reported in the sunitinib group (10%) versus 21 deaths in the placebo group (25%), which translated into a hazard ratio for death in favor of sunitinib (HR 0.41, 95% CI, 0.19–0.89, *p* = 0.02) at the data cutoff [8]. Median duration of follow-up for OS was 67.4 months in SUNNET during which time 59 patients (69%) randomized to placebo crossed over to sunitinib [27]. In the final analysis for OS, there was no statistically significant difference between the sunitinib arm and placebo arm (38.6 months vs. 29.1 months, respectively, HR 0.73; 95% CI, 0.50–1.06; *p* = 0.094) [27].

In NETTER-1, at the time of prespecified interim analysis of OS, 14 deaths had occurred in the ^177^Lu-DOTATATE arm, and 26 deaths had occurred in the high-dose octreotide LAR arm, which represented a 60% lower risk of death in the ^177^Lu-DOTATATE arm (HR 0.40, *p* = 0.004) [11]. Of the 231 randomized patients participating in NETTER-1, 101 of 117 patients (86%) in the ^177^Lu-DOTATATE arm and 99 of 114 patients (87%) in the high-dose octreotide LAR arm entered long-term follow-up [30]. Final OS analysis occurred 5 years after the last patient was randomized, following 142 deaths, with a median follow-up of approximately 76 months in both treatment arms [30]. During long-term follow-up, 41 of 114 patients (36%) in the high-dose octreotide LAR arm crossed over to receive subsequent RLT, 26 of whom did so within 24 months of randomization [30]. The median OS was 48.0 months (95% CI, 37.4–55.2) in the ^177^Lu-DOTATATE arm and 36.3 months (95% CI, 25.9–51.7) in the high-dose octreotide LAR arm (HR 0.84, 95% CI, 0.60–1.17, *p* = 0.30) [30].

Objective response rate was a secondary efficacy endpoint in the clinical trials. This measure is less useful than PFS for population-based treatment decision-making, given the low frequency and small range of responses observed, but measuring treatment response does have utility on an individual basis. Of the seven trials showing a benefit in time to tumor progression [6] or PFS [7,8,9,10,11,19] and reporting response data, objective response rates were lower with SSAs and everolimus (range, 2–5%) than with sunitinib (9%) [8], ^177^Lu-DOTATATE (18%) [11], or CAPTEM (33%) [19].

### 3.2. Safety

Both short- and long-term safety data are required for clinicians to make more accurate risk assessments when selecting initial and subsequent therapies for advanced NETs. Adverse events (AEs) associated with each therapy over long-term use were entirely consistent with those emanating from the primary clinical trials and no new safety signals were detected during follow-up (Table 2). There was asymmetry in the duration of time when AEs were collected in the primary clinical trials, due to the efficacy of the active interventions. The clinical trial safety data were not adjusted for time on treatment, which should be considered when interpreting AE incidence data.

In SUNNET, patients received sunitinib and placebo for a median duration of 4.6 months and 3.7 months, respectively [8]. The mean relative dose intensity (i.e., the ratio of administered doses to planned doses) was 91% in the sunitinib arm and 101% in the placebo arm. At least one dose interruption was reported more often in the sunitinib arm than in the placebo arm (30% vs. 12%), primarily because of AEs [8]. Grade ≥ 3 AEs occurring more frequently with sunitinib than placebo included diarrhea (5% vs. 2%), neutropenia (12% vs. 0%), hypertension (10% vs. 1%), stomatitis (4% vs. 0%), and thrombocytopenia (4% vs. 0%), although incidence of serious AEs was lower in the sunitinib arm (26.5% vs. 41.5%) [8].

In RADIANT-3 and RADIANT-4, median duration on everolimus was approximately double that of placebo [9,10]. The median relative dose intensities in these trials were 86–90% for everolimus and 97–100% for placebo. There was a high incidence of dose reduction or interruptions in the everolimus arms versus the placebo arms of RADIANT-3 (59% vs. 28%) [9] and RADIANT-4 (67% vs. 30%) [10]. The incidence of AEs resulting in permanent discontinuation was 20–30% for the everolimus arms [16]. Treatment-emergent AEs (TEAEs) such as stomatitis, rash, diarrhea, fatigue, edema, abdominal pain, nausea, fever, and headache account in large part for the everolimus safety profile [16].

The potential for acute, subacute, and long-term AEs with ^177^Lu-DOTATATE was characterized in the NETTER-1 study, which encompassed 5 years of patient follow up [11,30]. In the primary study, 77% of patients in the ^177^Lu-DOTATATE arm received all four infusions of ^177^Lu-DOTATATE, with 7% requiring a dose reduction. A numerically smaller proportion of patients in the ^177^Lu-DOTATATE arm than control arm had AEs resulting in premature withdrawal (6% vs. 9%) [11]. Transient grade 3 or 4 neutropenia, thrombocytopenia, and lymphopenia occurred in 1%, 2%, and 9%, respectively, of patients in the ^177^Lu-DOTATATE arm versus no patients in the control group [11]. During long-term follow up, the incidence of treatment-related serious AEs was 3% with none of these events occurring after the 5-year safety analysis cutoff [30]. Secondary hematologic malignancies and nephrotoxicity are AEs of interest for ^177^Lu-DOTATATE. Although two of 111 patients (1.8%) in the ^177^Lu-DOTATATE arm developed myelodysplastic syndrome, no patients developed myelodysplastic syndrome or acute myeloid leukemia during long-term follow-up [30]. Continued ^177^Lu-DOTATATE pharmacovigilance during real-world use is required to provide more clarity on the potential for myelodysplastic syndrome and acute myeloid leukemia, especially with respect to ^177^Lu-DOTATATE dose level, dose timing, and retreatment. There was no evidence of long-term nephrotoxicity in NETTER-1 [30].

### 3.3. Health-Related Quality of Life

Patient-reported outcome (PRO) data can reveal if the overall efficacy and safety profiles of therapy reflect patient experience and perceptions. The main PRO instrument used in PROMID [6], SUNNET [8], CLARINET [7], NETTER-1 [11], and SPINET [24] was the European Organization for Research and Treatment of Cancer Quality of Life Questionnaire C30 (EORTC QLQ-C30) and that in RADIANT-4 was the Functional Assessment of Cancer Therapy-General (FACT-G) [10]. PRO data were not reported in RADIANT-3 [9].

Only the efficacy and safety of ^177^Lu-DOTATATE (in NETTER-1) translated into demonstrable improvement in health-related quality of life (HRQoL), as evidenced by a substantially longer time to clinically meaningful deterioration in EORTC QLQ-C30 scores vs. high-dose octreotide LAR [43]. Octreotide LAR, lanreotide, sunitinib, and everolimus did not show statistically significant improvement or worsening relative to placebo on the EORTC QLQ-C30 or FACT-G (except for worsening of diarrhea with sunitinib), indicating that HRQoL is maintained in these patients to a certain extent [6,7,8,44].

## 4. Translating Clinical Research into Therapeutic Strategy

The NCCN Clinical Practice Guidelines in Oncology (NCCN Guidelines^®^) and North American Neuroendocrine Tumor Society guidelines for advanced NETs recommend use of SSAs as first-line systemic therapy followed by targeted therapy (everolimus or sunitinib), ^177^Lu-DOTATATE, or chemotherapy as later-line options depending on tumor type, grade, SSTR expression, distribution, and bulk of disease [5,13]. There are no robust categorized levels of evidence on optimal treatment strategies (i.e., order of administration, number of cycles, efficacy of combinations, and switching criteria) for scenarios often encountered in clinical practice. Currently available clinical data on treatment sequencing are restricted to a small number of retrospective studies [45].

Given the complexity of clinical decision-making under uncertainty and lack of clinical trial data, patients with advanced NETs should be referred to high-volume treatment centers. Treatment decisions should be joint efforts by the referring and high-volume treatment center and conducted by a multidisciplinary team (including the referring medical oncologist) that determines the patients’ initial treatment plan and any required changes to this plan when disease control is in jeopardy. The treatment plan should be based on disease-related factors and patient-related characteristics, such as comorbidities and patient preferences. How these factors and clinical trial data may guide treatment sequencing decisions in commonly encountered disease scenarios are summarized in Table 3. In real-world clinical practice, treatment decisions are highly individualized, but the best practices given in Table 3 can serve as useful starting points in clinical decision-making.

It should be acknowledged that there are several evidence gaps that hinder well-informed clinical decision-making in certain contexts. For example, there are currently limited head-to-head data for continuation of SSAs beyond disease progression, use of combination therapy, optimal treatment sequences, and potential for retreatment with ^177^Lu-DOTATATE. Despite these gaps, the current clinical data and our clinical experience have allowed us to provide a framework for treatment sequencing decisions across many different clinical scenarios, but more data are needed to fully support well-informed clinical decision-making for patients with NETs.

## 5. Conclusions

NETs have seen a significant advance in therapeutic drug development in the last decade. This has ushered us into a new era in which “how to sequence therapies” is a foremost challenge. New and emerging clinical trial data of agents for advanced NETs have demonstrated improved PFS versus control, with efficacy benefits preserved across clinically relevant patient subgroups. Long-term clinical trial follow-up has also evaluated the safety profile of each agent comprehensively, enabling clinicians to make more accurate risk assessments when selecting therapy. Clinical trial data and real-world data are now needed to meet the challenge of understanding how to use these valuable medicines optimally, and several studies are ongoing that may fill important knowledge gaps (Figure 3). In the meantime, continuing medical education and multidisciplinary team collaboration is important to ensure that clinicians continue to have the acumen and resources to make treatment sequencing decisions based on clinical trial data in conjunction with a full patient risk assessment.

## Figures and Tables

**Figure 1 cancers-14-05248-f001:**
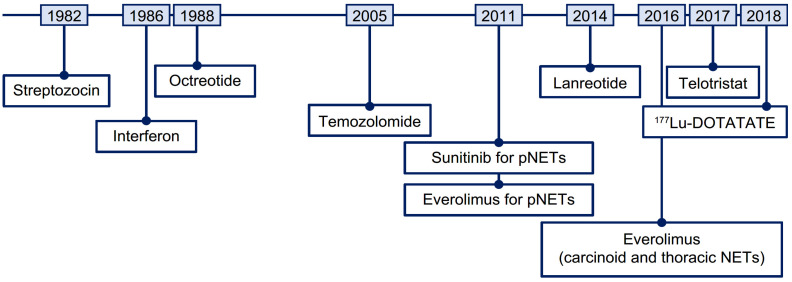
Timeline of therapies approved for the treatment of NETs. ^177^Lu, lutetium-177; NET, neuroendocrine tumor; pNET, pancreatic neuroendocrine tumor.

**Figure 2 cancers-14-05248-f002:**
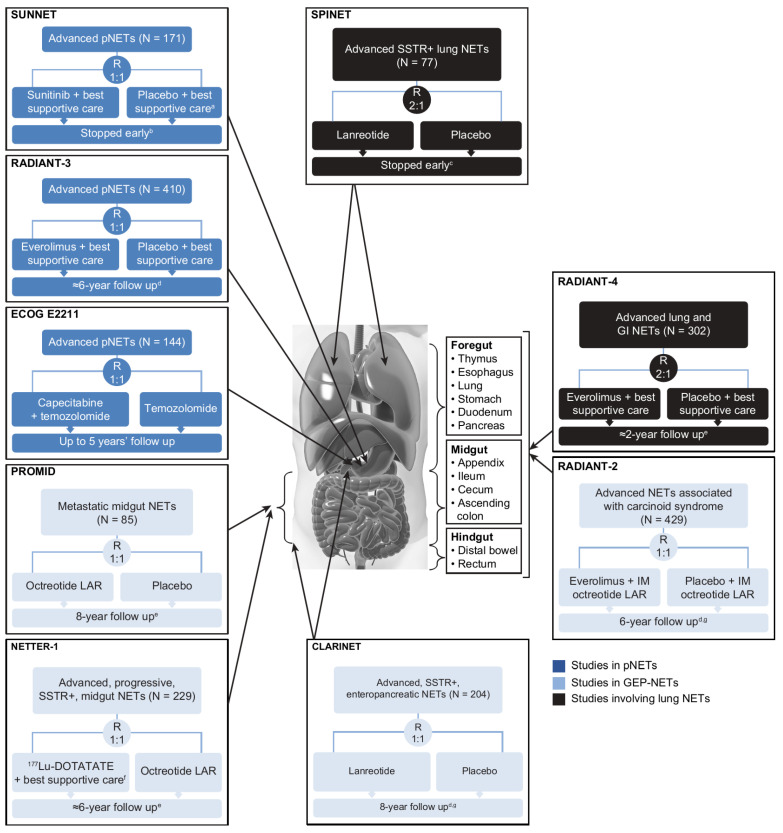
Randomized, controlled clinical trials conducted in the advanced NET setting. PROMID and CLARINET were the only trials involving primarily treatment-naïve populations; all other trials were beyond first-line therapy. ^a^ Patients could receive SSAs at the investigator’s discretion. ^b^ After the independent data and safety monitoring committee observed more serious adverse events and deaths in the placebo group as well as a difference in PFS favoring sunitinib. ^c^ Due to slow accrual. ^d^ Based on number of deaths. ^e^ Median. ^f^ Including IM octreotide LAR administered at a dose of 30 mg. ^g^ Including open-label extension phase. BID, twice daily; CLARINET, Controlled Study of Lanreotide Antiproliferative Response in Neuroendocrine Tumors; ECOG, Eastern Cooperative Oncology Group; GEP, gastroenteropancreatic; IM, intramuscular; LAR, long-acting repeatable; NETs, neuroendocrine tumors; NETTER-1, Neuroendocrine Tumors Therapy; PFS, progression-free survival; pNET, pancreatic NET; PROMID, placebo-controlled, double-blind, prospective, randomized study on the effect of octreotide LAR in the control of tumor growth in patients with metastatic neuroendocrine MID gut tumors; QD, once daily; RADIANT-3, RAD001 in Advanced Neuroendocrine Tumors, third trial; RECIST, Response Evaluation Criteria in Solid Tumors; SPINET, Efficacy and safety of lanreotide autogel/depot 120 mg vs. placebo in subjects with lung neuroendocrine tumors; SSAs, somatostatin analogs; SSTR, somatostatin receptor; SUNNET, Study of Sunitinib Compared to Placebo for Patients with Advanced Pancreatic Islet Cell Tumors; WHO PS, World Health Organization performance status.

**Figure 3 cancers-14-05248-f003:**
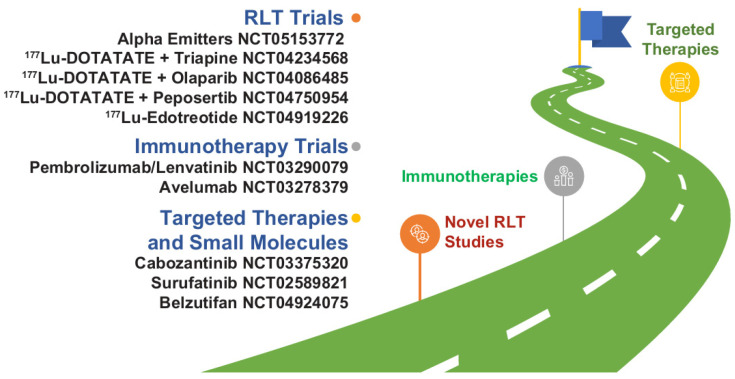
Novel therapy development in NETs as illustrated by select trials focusing on novel combinations and new agents or classes of drugs. Most of the agents listed are not commercially available in the United States for NETs. ^177^Lu, lutetium-177; NET, neuroendocrine tumor; RLT, radioligand therapy.

**Table 1 cancers-14-05248-t001:** Efficacy of systemic treatments for advanced NETs ^a^.

Study	No; Type of NET	Intervention, *N*	TTP/PFS			Final/Long-Term OS			Response, Active vs. Control, %
			Median, Months (95% CI)	HR (95% CI)	*p* Value	Median, Months (95% CI)	HR (95% CI)	*p* Value	
**First-line therapy**									
PROMID [6,29]	85; metastatic midgut NETs	Octreotide LAR 30 mg every 4 weeks, 42	TTP: 14.3 (11.0–28.8)	0.34 (0.20–0.59)	0.000072	84.7	0.83 (0.47–1.46)	0.51	CR: 0 PR: 2 vs. 2 SD: 67 vs. 37
		Placebo, 43	TTP: 6.0 (3.7–9.4)			83.7			
CLARINET [7,31]	204; advanced non-functional SSTR+ enteropancreatic NETs	Lanreotide 120 mg every 4 weeks, 101	PFS: not reached	0.47 (0.30–0.73)	<0.001	NR	NR	0.88	NR
		Placebo, 103	PFS: 18.0 (12.1–24.0)						
**Beyond first-line therapy**									
SUNNET [8,27]	171; advanced pNETs	Sunitinib 37.5 mg/day; ^b^ 86	PFS: 11.4	0.42 (0.26–0.66)	<0.001	38.6 (25.6–56.4)	0.73 (0.50–1.06)	0.094	ORR: 9 vs. 0 CR: 2 vs. 0 PR: 7 vs. 0 SD: 63 vs. 60
		Placebo; ^b^ 85	PFS: 5.5			29.1 (16.4–36.8)			
RADIANT-2 [23,32]	429; advanced NETs associated with carcinoid syndrome	Everolimus 10 mg/day; ^c^ 216	PFS: 16.4 (13.7–21.2)	0.77 (0.59–1.00)	0.026	29.2 (23.8–35.9)	1.17 (0.92–1.49)	NR	CR: 0 PR: 2 vs. 2 SD: 84 vs. 81
		Placebo; ^c^ 213	PFS: 11.3 (8.4–14.6)			35.2 (30.0–44.7)			
RADIANT-3 [9,28]	410; advanced pNETs	Everolimus 10 mg/day, 207	PFS: 11.0 (8.4–13.9)	0.35 (0.27–0.45)	<0.001	44.0 (35.6–51.8)	0.94 (0.73–1.20)	0.3	CR: 0 PR: 5 vs. 2 SD: 73 vs. 51
		Placebo, 203	PFS: 4.6 (3.1–5.4)			37.7 (29.1–45.8)			
RADIANT-4 [10]	302; advanced, non-functional lung and GI NETs	Everolimus 10 mg/day, 205	PFS: 11.0 (9.2–13.3)	0.48 (0.35–0.67)	<0.00001	27.3	0.64 (0.40–1.05)	0.037	CR: 0 PR: 2 vs. 1 SD: 81 vs. 64
		Placebo, 97	PFS: 3.9 (3.6–7.4)			NA			
NETTER-1 [11,30,33,34]	229; advanced SSTR+ midgut NETs progressing on octreotide LAR (20 to 30 mg)	^177^Lu-DOTATATE 7.4 GBq every 8 weeks, ^d^ 116	PFS: not reached	0.18 (0.11–0.29)	<0.0001	48.0 (37.4–55.2)	0.84 (0.60–1.17)	0.3	ORR: 18 vs. 3 CR: 1 vs. 0 PR: 17 vs. 3
		Octreotide LAR 60 mg every 4 weeks, 113	PFS: 8.5 (5.8–9.1)			36.3 (25.9–51.7)			
ECOG E2211 [19]	144; advanced pNETs	CAP 750 mg/m^2^ BID (days 1–14) + TEM 200 mg/m^2^ QD (days 10–14); 72	PFS: 22.7	0.58	0.022	58.7	0.82	0.42	NR
		TEM 200 mg/m^2^ QD (days 1–5); 72	PFS: 14.4			53.8			
SPINET [24]	77; advanced SSTR+ lung NETs	Lanreotide 120 mg every 4 weeks; 51	PFS: 16.6 (11.3–21.9) ^e^	0.90 (0.46–1.88)	0.769	NR	NR	NR	ORR: 14 vs. 0
		Placebo; 26	13.6 (8.3–NE) ^e^						

^a^ Direct comparison between these trials is not possible because of variations in patient selection and study methodology. ^b^ Patients could receive SSAs at the investigator’s discretion. ^c^ In conjunction with IM octreotide 30 mg LAR every 28 days. ^d^ Including IM octreotide LAR administered at a dose of 30 mg. ^e^ Primary endpoint changed on trial to PFS in the double-blind and open-label lanreotide phases in patients initially randomized to lanreotide. BID, twice daily; CAP, capecitabine; CLARINET, Controlled Study of Lanreotide Antiproliferative Response in Neuroendocrine Tumors; ECOG, Eastern Cooperative Oncology Group; GI, gastrointestinal; LAR, long-acting repeatable; NA, not available; NETs, neuroendocrine tumors; NETTER-1, Neuroendocrine Tumors Therapy; NE, not estimable; NR, not reported; ORR, objective response rate; OS, overall survival (observed all-cause); PFS, progression-free survival; PROMID, placebo-controlled, double-blind, prospective, randomized study on the effect of octreotide LAR in the control of tumor growth in patients with metastatic neuroendocrine MID gut tumors; QD, once daily; RADIANT, RAD001 in Advanced Neuroendocrine Tumors; SPINET, Efficacy and safety of lanreotide autogel/depot 120 mg vs. placebo in subjects with lung neuroendocrine tumors; SSAs, somatostatin analogs; SSTR, somatostatin receptor; TEM, temozolomide; TTP, time to tumor progression.

**Table 2 cancers-14-05248-t002:** Tolerability, safety, and monitoring issues of therapeutics for advanced NETs.

Agent	Common AEs	Safety Issues	Long-Term Safety Considerations
Octreotide LAR [35]	Incidence > 20%	Cholelithiasis and complications of cholelithiasis, hypoglycemia or hyperglycemia, hypothyroidism, cardiac dysfunction	Six years of octreotide use in patients with acromegaly did not reveal any new safety signals with its prolonged use [36]
	Back pain, fatigue, headache, abdominal pain, nausea, dizziness		
Lanreotide [37]	Incidence > 10%	Cholelithiasis and complications of cholelithiasis, hypoglycemia or hyperglycemia, hypothyroidism, bradycardia	Incidences of AEs and treatment-related AEs were lower in the open-label extension study than in the core study [31]
	Abdominal pain, musculoskeletal pain, vomiting, headache, injection site reaction, hyperglycemia, hypertension, cholelithiasis		
Sunitinib [17,38]	Incidence ≥ 25% ^a^	Hepatotoxicity, cardiovascular events, QT interval prolongation and torsade de pointes, hypertension, hemorrhagic events, tumor lysis syndrome, thrombotic microangiopathy, proteinuria, dermatologic toxicities, reversible posterior leukoencephalopathy syndrome, thyroid dysfunction, hypoglycemia, osteonecrosis of the jaw, impaired wound healing, embryo-fetal toxicity	In patients with renal cell carcinoma, prolonged treatment was not associated with new adverse events or increased yearly incidence
	Fatigue/asthenia, diarrhea, mucositis/stomatitis, nausea, decreased appetite/anorexia, vomiting, abdominal pain, hand-foot syndrome, hypertension, bleeding events, dysgeusia/altered taste, dyspepsia, and thrombocytopenia		
Everolimus [16]	Incidence ≥ 30% ^a^	Non-infectious pneumonitis, infections, severe hypersensitivity reactions, angioedema, stomatitis, renal failure, impaired wound healing, metabolic disorders, myelosuppression, risk of infection or reduced immune response with vaccination, radiation sensitization and radiation recall, embryo-fetal toxicity	Safety data from final OS analysis were consistent with the previously reported safety profile [28]
	Stomatitis, infections, rash, fatigue, diarrhea, edema, abdominal pain, nausea, fever, asthenia, cough, headache, decreased appetite		
^177^Lu-DOTATATE [18,30]	Grade 3–4 AEs (≥4% with a higher incidence in ^177^Lu-DOTATATE arm)	Risk from radiation exposure, myelosuppression: secondary myelodysplastic syndrome and leukemia, renal toxicity, hepatotoxicity, neuroendocrine hormonal crisis, embryo-fetal toxicity, risk of infertility	Long-term safety data were consistent with current practice regarding low rates of myelodysplastic syndrome (1.8%; 2 of 111 patients), acute leukemia, and nephrotoxicity and previously published meta-analysis data [30,39] No new myelodysplastic syndrome or acute leukemia with 5-year follow-up [30]
	Lymphopenia, increased GGT, vomiting, nausea, increased AST, increased ALT, hyperglycemia, hypokalemia		
CAPTEM [40,41,42]	Grade 3 to 4 toxicities	Cardiotoxicity, myelosuppression, coagulopathy, opportunistic infections, diarrhea, dehydration and renal failure, myopathy, severe toxicity due to dihydropyrimidine dehydrogenase deficiency, mucocutaneous and dermatologic toxicity, hyperbilirubinemia	Myelodysplastic syndrome Secondary malignancy
	Thrombocytopenia (3.4%), neutropenia (0.7%), lymphopenia (0.7%), anemia (0.6%), mucositis (0.6%), fatigue (0.5%), diarrhea (0.5%), nausea (0.4%), and transaminase elevation (0.1%)		

^a^ Incidence reported across various tumor types. AE, adverse event; ALT, alanine aminotransferase; AST, aspartate aminotransferase; CAPTEM, capecitabine and temozolomide; GGT, gamma-glutamyl transferase.

**Table 3 cancers-14-05248-t003:** Author opinion on treatment sequencing in advanced NETs.

Scenario	Multidisciplinary Perspective on Treatment	Challenges and Considerations
Asymptomatic, liver metastasis ^a^	Grade 1 disease: Asymptomatic patients (especially with low bulk disease) can be safely observed with interval scans for tumor growth rate, and systemic therapy can be instituted at the time of progression; multidisciplinary team review can help determine if resection is a treatment option to cure the disease or prolong OS/PFS [46]; Grade ≥ 2 disease: Consider initiation of SSAs (especially when Ki67 > 10%) after informed discussion with the patient	Frequency and timing of interval scans are not standardized; however, it is reasonable to get cross-sectional imaging every 3–6 months; Determination of the optimal time to start SSAs must be based on volume of disease, disease stability, and patient preference as it is difficult to discontinue these agents once started
High-volume and/or symptomatic liver-dominant disease	Discuss treatment options in a multidisciplinary setting; Based on the volume of disease, consider local therapy with surgical resection and/or liver-directed therapy (embolic therapy preferred); transplant is also an option Often, systemic treatment, especially with SSAs, can be considered in addition to local therapy in bulky disease	Transplant criteria are very selective and are likely only applicable to the occasional patient
Local and loco-regional disease (site agnostic) ^a^	Curative intent surgery in surgical candidates; For nonoperative candidates, the algorithm for metastatic disease can be followed	
Bronchial NETs ^a^	SSAs are the preferred front-line treatment for metastatic progressive SSTR+ bronchial NETs; Everolimus is an excellent second-line therapy based on RADIANT-4 data [10]. RLT can be considered in SSTR+ bronchial NETs refractory to SSAs CAPTEM can be considered in bronchial NETs refractory to standard treatment, especially atypical NETs and SSTR negative bronchial NETs [47]; Observation may be considered in certain patients (e.g., scattered lung nodules that are stable); Participation in relevant clinical trials is highly encouraged	Treatment is dependent on several factors including burden of SSTR+ disease and whether there is an immediate need for response; Role of Ki-67 for bronchial NETs is controversial; The ALLIANCE trial (NCT04665739) will determine if everolimus or ^177^Lu-DOTATATE is preferred DOTATATE/DOTATOC PET-CT imaging is strongly preferred to assess SSTR+ bronchial NETs (over ^111^In-pentetreotide scintigraphy due to significant lower sensitivity, particularly in bronchial NETs)
Low-volume or asymptomatic bronchial NETs	Consider observation first; If treatment is desired, consider local therapy (SBRT) and SSAs	
High-volume and symptomatic bronchial NETs	^177^Lu-DOTATATE preferred in SSTR+ patients. Everolimus can also be considered; CAPTEM should be reserved for very symptomatic patients requiring a rapid response, especially higher-grade tumors (atypical carcinoid)	Although the role of Ki-67 scores for bronchial NETs is controversial, consider obtaining a score for patients with aggressive disease; The factors influencing the decision of when and when not to use platinum-based chemotherapy (e.g., based on Ki-67 score) are not well defined; Data are emerging for ipilimumab/nivolumab in this setting and can be an option in the refractory setting
Low-volume and asymptomatic pNETs ^a^	Therapy should be based on safety and efficacy data. A therapy associated with a high ORR is not a priority. Therefore, consider SSA > everolimus, sunitinib, or ^177^Lu-DOTATATE > CAPTEM as a possible rank order of therapies	There is a lack of consensus on optimal treatment sequencing in this setting. Patient preference is an especially important consideration in this scenario given the lack of comparative efficacy/safety data
pNETs progressing despite SSA therapy	Consider everolimus, sunitinib, ^177^Lu-DOTATATE (in select populations, as it is likely better tolerated than everolimus/sunitinib), or CAPTEM (choosing among these treatments depends on patient comorbidities and side-effect profiles [see Table 2]; CAPTEM or ^177^Lu-DOTATATE may be especially suitable in tumors with a faster growth rate and more bulky disease if a response is needed	Increase in SSA dose intensity can be considered for patients who are not able to receive other treatments
High-volume and symptomatic pNETs	Consider CAPTEM first line for efficient cytoreduction; Alternatively, RLT can be considered for SSTR+ disease when disease shrinkage is desired. RLT is also ideal for widespread bony metastatic disease; Everolimus is a good second-line option for low-volume disease; Sunitinib should be reserved for third-line and later treatment owing to its side-effect profile and availability of more tolerable alternative options; SSAs are usually added; however, ORRs with SSAs are modest (<5%); For liver-dominant disease, consider liver-directed therapy (which can be sequenced with systemic therapy)	Metastatic pNETs with high-volume symptomatic disease or risk of rapid progression require an objective response or tumor shrinkage; The A022001 trial (NCT05247905) will determine if CAPTEM or ^177^Lu-DOTATATE is preferred in metastatic pNETs; In patients with liver-dominant disease who have undergone pancreaticoduodenectomy, risk of liver-directed therapy may outweigh benefit, given the high risk of infection from enteric seeding of the biliary tract
Low-volume or asymptomatic midgut NETs ^a^	Consider frontline SSAs; Upon disease progression, consider an increase in SSA dose (select patients with slow/minimal radiographic progression), ^177^Lu-DOTATATE, or everolimus	
Midgut NETs with low disease burden on SSA	Consider loco-regional therapies like ablation or hepatic artery embolization for liver-dominant progressive disease; Surgical debulking for low-risk patients should be considered after appropriate discussion at a high-volume treatment center; Diffuse hepatic and extrahepatic progression warrant treatment with everolimus or ^177^Lu-DOTATATE; ^177^Lu-DOTATATE is preferred over everolimus in symptomatic patients due to superior ORR/cytoreductive capacity	
Functional midgut NETs	Consider SSA > ^177^Lu-DOTATATE > everolimus (given negative Phase III data from RADIANT 2) [23] as a possible rank order of therapies	For functional NETs, SSAs should be continued beyond progression for symptom control
High volume or symptomatic midgut NETs ^a^	RLT can be rapidly introduced after starting SSAs if the patient continues to have symptomatic disease after starting an SSA; Liver-directed therapy and surgical debulking (e.g., for impending bowel obstruction) can be considered; Everolimus is a good third-line option and can also be considered as a second-line option for patients with low-volume asymptomatic disease	The NETTER-2 study (NCT03972488) is comparing ^177^Lu-DOTATATE plus octreotide LAR with high-dose octreotide LAR for frontline treatment of GEP-NET patients with high proliferation rate tumors (G2 and G3); Addition of steroids to ^177^Lu-DOTATATE is becoming increasingly common to reduce inflammatory responses
Progressing after RLT	Consider other established agents that have not been used; Potential for limited dose (200 mCi × 2 or 100 mCi × 4) RLT retreatment in appropriate patients; Referral to high-volume centers for assessment and participation in clinical trials are highly encouraged	A CCTG-SWOG RCT (NETRETREAT) is in development to prospectively study safety and efficacy of RLT retreatment; RLT retreatment does not apply to patients that have primary progression after RLT; Insurance coverage may be challenging

^a^ See Appendix A for example imaging of the clinical scenario. ^111^In, indium-111; ^177^Lu, lutetium-177; CAPTEM, capecitabine and temozolomide; CCTG, Canadian Clinical Trials Group; GEP-NET, gastroenteropancreatic neuroendocrine tumor; LAR, long-acting repeatable; NET, neuroendocrine tumor; ORR, objective response rate; OS, overall survival; PET-CT, positron emission tomography-computed tomography; PFS, progression-free survival; pNET, pancreatic neuroendocrine tumor; RCT, randomized controlled trial; RLT, radioligand therapy; SBRT, stereotactic body radiation therapy; SSA, somatostatin analog; SSTR, somatostatin receptor; SWOG, Southwest Oncology Group.

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
