# Peer review of "Treatment Sequencing Strategies in Advanced Neuroendocrine Tumors: A Review"

_cancers, 2022, doi:10.3390/cancers14215248_

Round 1
Reviewer 1 Report
Chauhan et al claim to review the highest quality clinical trials data to address the point of sequencing treatment for NETs. Most of the review is that, a review. AS there are no data for sequencing, there is little addressed herein. Comments on sequencing would be a commentary rather than a review. The sequencing of SSAs prior to anything else in otherwise symptomatic but not at risk patients is well known and covered. The title should be revised to reflect this. There is little in the text to address what to do otherwise, absent the issues of speed of tumor growth, end organ risk.
Specific comments:
Figure 1 should be adjusted to linear scale to show the time line more accurately.
Caution should be raised about stating highest quality literature unless using formalized validated metrics. Indicate if systematic review criteria were used and if not, how studies were selected. Were all randomized? Minimum size? Etc.
Provide the results before the discussion.
Time to tumor progression and progression-free survival are the same thing. It is unclear what point was being made about that in the first paragraph of the discussion. Harmonize throughout.
While Figure 2 is cute, it would be more user friendly if it were done in table form, which is the case for table 1. Duplication is not necessary. Modify table 1 if needed. Many of the points in the footnotes to table 1 warrant statement in the text, especially the acknowledgement that cross trial comparisions are fraught, given the lack of harmonized eligibility and interventions.
Section 3.1 begins with a comment on PFS, but then gets into ORR data. Make clear what the primary endpoint of the trial and thus the valued data are. Secondary endpoints, such as ORR should be discussed separately, especially as they are less useful for decision making given the low frequency and small range.
Make sure that every use of the word “survival” is qualified by progression-free or overall so that the reader knows to what the authors refer.
Be sure to qualify what are predetermined comparisons v. post-hoc evaluations. Most elements on a tree plot are demographic elements which are not statistically prespecified analytical variables. Do not lump the quality data with the post hoc data without acknowledgement.
Were all OS data unspecified or disease-specific?
Safety is an important element. The question about the real frequency of AML/MDS is still open. That it can happen within 2 years of treatment is unusual and notable. It is an important point to raise and consider whether there are differential risks with different radionuclides, exposures (dose and time) as retreatment with PRRT is happening world wide absent data on activity or safety. Including data on Y90PRRT, even if not from randomized studies may be important.
Who are the sources for your expert opinion in Table 3? What is the justification for this? Is it in agreement with published guidelines developed with formalized metrics? Why reiterate? Reconsider a cartoon showing recommendations, to remove the suggestion that these were generated in the rigorous fashion done by guideline groups such as ASCO and NCCN. Alternatively, consider situations where there is controversy in what to do and discuss those in the context of the multiple variables and unknowns. Address the unknowns in these cases and others in more detail.
There are more gaps in knowledge than there are data to support recommendations. This should be more carefully addressed.
Remove the title of improved efficacy and lower toxicity from the cartoon of studies in progress. It implies that is the outcome. There are limited and poor data on IO, mostly single arm studies, and no harmonization for use. Many of the agents listed are not in the US armamentarium and should be so qualified.
The use and value of the supplement is unclear.
Editorial: review grammar and correct tenses.
Reviewer 2 Report
Thank you for allowing me to review your very comprehensive review on "Evidence-Based Treatment Sequencing Strategies in Advanced Neuroendocrine Tumors". This review is well organized with appropriate references to critical trials.
Treatment algorithm suggestions are well defined. Some minor suggestions to consider.
Would consider enlarging Figure 2 as text is very difficult to read (either enlarge figures or limit text within boxes)
Page 10/15 Scenario listing "asymptomatic liver disease" - when mentioning low bulk disease, can consider resection in these settings if it will be curative
Additionally, there is retrospective data suggesting improved PFS with >70% debulking.
Would consider stating that this could be an option or at least that multid review is warranted to discuss treatment options
For high volume disease - would include transplant as an option in addition with surgery - though criteria are very selective, it may be applicable to an occasional patient
For high volume and symptomatic PNETs, when mentioning liver dominant disease and liver direct therapy, may be worth alluding to the fact that post pancreaticoduodenectomy (if patients were previously treated), risk of liver directed therapy may outweigh benefit given high risk of infection from enteric seeding of biliary tract
Would re check the references as there is something that has gone awry with the numbering starting with ref number 14
